# Generative Cleaning Networks with Quantized Nonlinear Transform for Deep Neural Network Defense

## Abstract

Effective defense of deep neural networks against adversarial attacks remains a challenging problem, especially under white-box attacks. In this paper, we develop a new generative cleaning network with quantized nonlinear transform for effective defense of deep neural networks. The generative cleaning network, equipped with a trainable quantized nonlinear transform block, is able to destroy the sophisticated noise pattern of adversarial attacks and recover the original image content. The generative cleaning network and attack detector network are jointly trained using adversarial learning to minimize both perceptual loss and adversarial loss. Our extensive experimental results demonstrate that our approach outperforms the state-of-art methods by large margins in both white-box and black-box attacks. For example, it improves the classification accuracy for white-box attacks upon the second best method by more than 40% on the SVHN dataset and more than 20% on the challenging CIFAR-10 dataset.

## 1 Introduction

Recent research has shown that deep neural networks are sensitive to adversarial attacks (Szegedy et al., 2013). Very small changes of the input image can fool the state-of-art classifier with very high success probabilities. During the past few years, a number of methods have been proposed to construct adversarial samples to attack the deep neural networks, including fast gradient sign (FGS) method (Goodfellow et al., 2014b), Jacobian-based saliency map attack (J-BSMA) (Papernot et al., 2016a), and projected gradient descent (PGD) attack (Kurakin et al., 2016; Madry et al., 2018). It has also been demonstrated that different classifiers can be attacked by the same adversarial perturbation (Szegedy et al., 2013). The fragility of deep neural networks and the availability of these powerful attacking methods present an urgent need for effective defense methods. During the past few years, a number of deep neural network defense methods have been developed, including adversarial training (Kurakin et al., 2016; Szegedy et al., 2013), defensive distillation (Papernot et al., 2016b; Carlini & Wagner, 2016; Papernot & McDaniel, 2016), Magnet (Meng & Chen, 2017) and featuring squeezing (He et al., 2017; Xu et al., 2017). It has been recognized that these methods suffer from significant performance degradation under strong attacks, especially white-box attacks with large magnitude and iterations (Samangouei et al., 2018).

In this work, we explore a new approach to defend various attacks by developing a generative cleaning network with quantized nonlinear transform. We recognize that the attack noise is not random and has sophisticated patterns. The attackers often generate noise patterns by exploring the specific network architecture or classification behavior of the target deep neural network so that the small noise at the input layer can accumulate along the network inference layers, finally exceed the decision threshold at the output layer, and result in false decision. On the other hand, we know a well-trained deep neural networks are robust to random noise (Arjovsky et al., 2017), such as Gaussian noise. Therefore, the key issue in network defense is to randomize or destroy the sophisticated pattern of the attack noise while preserving the original image content.

Motivated by this observation, we design a new generative cleaning network with quantized nonlinear transform to first destroy the sophisticated noise patterns of adversarial attacks and then recover the original image content damaged during this nonlinear transform. We also construct a detector

network which serves as the dual network for the target classifier to be defended. The generative cleaning network and detector network are jointly trained using adversarial learning so that the detector network cannot detect the existence of attack noise pattern in the images recovered by the generative cleaning network. Our extensive experimental results demonstrate that our approach outperforms the state-of-art methods by large margins in both white-box and black-box attacks. It significantly improves the classification accuracy for white-box PGD attacks upon the second best method by more than 40% on the SVHN dataset from 46.90% to 93.80%, and more than 20% on the challenging CIFAR-10 dataset from 60.15% to 86.05%.

The **major contributions** of this work can be summarized as follows. (1) We have proposed a new approach for deep neural network defense by developing a unique generative cleaning network with quantized nonlinear transform. (2) We have formulated the problem of destroying the noise patterns of adversarial attacks and reconstructing original image content into generative adversarial network design and training which considers both perceptual loss and adversarial loss. (3) Our new method has significantly improved the performance of the state-of-the-art methods in the literature under a wide variety of attacks.

The rest of this paper is organized as follows. Section 2 reviews related work. The proposed method is presented in Section 3. Experimental results and performance comparison with existing methods are provided in Section 4. Section 5 concludes the paper.

## 2 RELATED WORK

In this section, we review related work on adversarial attack and network defense methods.

**(A) Attack methods.** Attack methods can be divided into two threat models: white-box attacks and black-box attacks. The white-box attacker has full access to the classifier network parameters, network architecture, and weights. The black-box attacker has no knowledge of or access to the target network. For white-box attack, a simple and fast approach called *Fast Gradient Sign (FGS)* method has been developed by Goodfellow et al. (2014b) using error back propagation to directly modify the original image. Basic Iterative Method (BIM) is an improved version of the FGS method. Carlini & Wagner (2016) designed an optimization-based attack method, called *Carlini-Wagner (C&W) attack*, which is able to fool the target network with the smallest perturbation. Xiao et al. (2018) trained a generative adversarial network (GAN) (Goodfellow et al., 2014a) to generate perturbations. Kannan et al. (2018) found that the *Projected Gradient Descent (PGD)* is the strongest among all attack methods. It can be viewed as a multi-step variant of $FGS^k$ (Madry et al., 2018). Athalye et al. (2018) introduced a method, called Backward Pass Differentiable Approximation (BPDA), to attack networks where gradients are not available. It is able to successfully attack all existing state-of-the-arts defense methods. For black-box attacks, the attacker has no knowledge about target classifier. Papernot et al. (2017) introduced the first approach for black-box attack using a substitute model. Dong et al. (2018) proposed a momentum-based iterative algorithms to improve the transferability of adversarial examples. Xie et al. (2018c) boosted the transferability of adversarial examples by creating diverse input patterns.

**(B) Defense methods** Several approaches have recently been proposed for defending both white-box attacks and black-box attacks. Adversarial training defends various attacks by training the target model with adversarial examples (Szegedy et al., 2013; Goodfellow et al., 2014b). Madry et al. (2018) suggested that training with adversarial examples generated by PGD improves the robustness. Meng & Chen (2017) proposed a method, called MagNet, which detects the perturbations and then reshape them according to the difference between clean and adversarial examples. Recently, there are several defense methods based on GANs have been developed. Samangouei et al. (2018) projected the adversarial examples into a trained generative adversarial network (GAN) to approximate the input using generated clean image with multiple iterations. Recently, some defense methods have been developed based on input transformations. Guo et al. (2018) proposed several input transformations to defend the adversarial examples, including image cropping and re-scaling, bit-depth reduction, and JPEG compression. Xie et al. (2018a) proposed to defend against adversarial attacks by adding a randomization layer, which randomly re-scales the image and then randomly zero-pads the image. Jia et al. (2019) proposed an image compression framework to defend adversarial examples, called *ComDefend*. Xie et al. (2018b) introduced a feature denoising method for defending PGD white-box attacks.

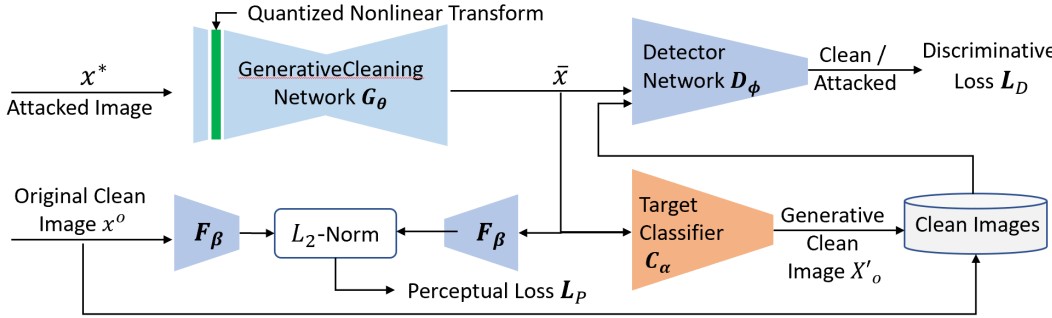

Figure 1: Overview of our generative cleaning network method for deep neural network defense.

Our proposed defense method is also related to GANs and image transformations. But, compared to existing methods, our method is unique in the following aspects: (1) We introduce a special layer called quantized nonlinear transform, into the generative cleaning network to destroy the sophisticated noise pattern of adversarial attacks. (2) Unlike the GAN-based methods in (Wang & Yu, 2019; Xiao et al., 2018) which aim to approximate input noise image using images generated by the GAN over multiple iterations, our generative cleaning network aims to reconstruct the image content damaged by quantized nonlinear transform. (3) Our method does not need to modify the target network to be protected.

## 3 THE PROPOSED DEFENSE METHOD

In this section, we present our proposed generative cleaning network method for effective deep neural network defense. For convenience, we refer to our proposed method by *GCLN*.

### 3.1 METHOD OVERVIEW

Figure 1 provides an overview of the proposed method. The attacked image $x^*$ is fed into the generative cleaning network $\mathbf{G}_\theta$. The network has a special layer, called quantized nonlinear transform, to destroy the noise pattern of the adversarial attack in the input image. The generative cleaning network aims to recover the original image content and produce a recovered image $\bar{x}$. This recovered image $\bar{x}$ will be passed to the target classifier $\mathbf{C}_\alpha$ for image classification or recognition. To successfully learn the generative cleaning network $\mathbf{G}_\theta$, we construct a detector network $\mathbf{D}_\phi$, which serves as the dual network for the target classifier network $\mathbf{C}_\alpha$. The task of $\mathbf{D}_\phi$ is to determine if the input image is clean or being attacked. In our proposed method, the generative cleaning network $\mathbf{G}_\theta$ and the detector network $\mathbf{D}_\phi$ are jointly trained through adversarial learning: the $\mathbf{G}_\theta$ network is trying to recover the image $\hat{x}$ so that $\mathbf{D}_\phi$ cannot detect any attack noise in it. In the following sections, we will explain the proposed method in more detail.

### 3.2 QUANTIZED NONLINEAR TRANSFORM LAYER IN THE GENERATIVE CLEANING NETWORK

During the generative cleaning network design, We incorporate one special layer into the network, called quantized nonlinear transform. This transform aims to disturb and partially destroy the sophisticated pattern of the attack noise. In this work, we propose to construct such a transform using a linear transform $T$, followed by a quantizer $Q$ and an inverse transform $T^{-1}$. For the linear transform, we can use the discrete cosine transform (DCT) (Ahmed et al., 1974) which has been in JPEG image compression (Wallace, 1992). Specifically, we partition the input image into blocks of $M \times M$. The original image block is denoted by $\mathbf{X}_B^* = [x_{nk}^*]_{1 \le n,k \le M}$. The output block $\hat{\mathbf{X}}_B = [\hat{x}_{ij}]_{1 \le i,j \le M}$ after DCT transform is given by

$$\hat{x}_{ij} = \frac{1}{4} C_i C_j \sum_{n=0}^{M-1} \sum_{k=0}^{M-1} x_{nk} \, \cos(i\pi \frac{2n+1}{2M}) \cos(j\pi \frac{2k+1}{2M}), \qquad (1)$$

with $C_i = 1/\sqrt{2}$ for $i = 0$, and $C_i = 1$ for $i \neq 0$. After transform, we will quantize the transform coefficient $\hat{x}_{ij}$ as follows

$$RQ(\hat{x}_{ij}) = Round\left(\frac{\hat{x}_{ij}}{q}\right) \times q, \tag{2}$$

where $q$ is the quantization parameter. Certainly, this DCT transform can be replaced with other invertible transform, such as discrete wavelet transform (Daubechies, 1990). During network training, this special quantized nonlinear transform layer is implemented in the same way as the pooling layers in existing deep neural networks and included into the training process of the whole generative cleaning network.

### 3.3 ADVERSARIAL TRAINING FOR GENERATIVE CLEANING NETWORKS

In our defense method design, the generative cleaning network $\mathbf{G}_\theta$ and the detector network $\mathbf{D}_\phi$ are trained against each other, just like the existing generative adversarial networks (GAN). $\mathbf{D}_\phi$ is a binary classifier to detect if the input image is clean or not. During the initial phase of training, $\mathbf{D}_\phi$ is trained with the clean images and their attacked versions generated by existing attack methods. It should be noted that, when training $\mathbf{D}_\phi$, we do not need to know the model inside the target network $\mathbf{C}_\alpha$.

The goal of the generative cleaning network $\mathbf{G}_\theta$ is two-fold: (1) first, it needs to successfully remove the residual attack noise so that the noise cannot be detected by the detector network $\mathbf{D}_\phi$. (2) Second, it needs to make sure that the original image content is largely recovered. To achieve the above two goals, we formulate the following generative loss function for training the generative cleaning network $\mathbf{G}_\theta$

$$\mathbf{L_G} = \lambda\mathbf{L}_P + (1 - \lambda)\mathbf{L}_A, \tag{3}$$

where $\mathbf{L}_P$ is perceptual loss and $\mathbf{L}_A$ is the adversarial loss. $\lambda$ is a weighting parameter. In our experiments, we set it to be 0.5. To define the perceptual loss, the $L_2$-norm between the recovered image $\bar{x}$ and the original image $x^o$ is often used (Johnson et al., 2016). In this work, we observe that the small adversarial perturbation often leads to very substantial noise in the feature map of the network (Xie et al., 2018b). Motivated by this, we use a pre-trained VGG-19 network, denoted by $\mathbf{F}_\beta$ to generate visual features for the recovered image $\bar{x}$ and the original image $x^o$, and use their feature difference as the perceptual loss $\mathbf{L}_P$. Specifically,

$$\mathbf{L}_P = ||\mathbf{F}_\beta(x^o) - \mathbf{F}_\beta(\mathbf{G}_\theta(\hat{x}))||_2, \tag{4}$$

The adversarial loss $\mathbf{L}_A$ aims to train $\mathbf{G}_\theta$ so that the recovered images will be detected as clean by the detector network $\mathbf{D}_\phi$. It is formulated as

$$\mathbf{L}_A = \mathbb{E}_{x^* \in \Omega^*}\Phi[\mathbf{D}_\phi(\mathbf{G}_\theta(x^*)), \mathbf{I}_{clean}]. \tag{5}$$

Here, $\Phi[\cdot, \cdot]$ represents the cross-entropy between the output generated by the generative network and the target label $\mathbf{I}_{clean}$ for clean images. We train our discriminative network $\mathbf{D}_\phi$, along with the generative cleaning network $\mathbf{G}_\theta$, to optimize the following min-max loss function:

$$\min_{\mathbf{G}_\theta} \max_{\mathbf{D}_\phi} \{\mathbb{E}_{x^o \in \Omega^o}[\log \mathbf{D}_\phi(x^o)] + \mathbb{E}_{x^* \in \Omega^*}[\log(1 - \mathbf{D}_\phi(\mathbf{G}_\theta(x^*)))]\}. \tag{6}$$

Here, $\Omega^o$ and $\Omega^*$ represent the clean and attacked images of the training dataset. The goal of generative model $\mathbf{G}_\theta$ is to fool the discriminator $\mathbf{D}_\phi$ that is trained to distinguish adversarial images from clean images. With this framework, our generator learns to recover images that are highly similar to clean images and difficult to be detected by $\mathbf{D}_\phi$. The detector network $\mathbf{D}_\phi$ acts as a dual network for the original classifier $\mathbf{C}_\alpha$. Cascaded with the generative cleaning network $\mathbf{G}_\phi$, it will guide the training of $\mathbf{G}_\phi$ using back propagation of gradients from its own network, aiming to minimize the above loss function. In our design, during the adversarial learning process, the target classifier $\mathbf{C}_\alpha$ is called to determine if the recovered image $\bar{x}$ is clean or not, as illustrated in Figure 1. If it is clean, it is added back into the clean training sample set $\Omega^o$ on the fly to enhance the learning process.

## 4 EXPERIMENTAL RESULTS

In this section, we implement and evaluate our GCLN defense method and compare its performance with state-of-the-art defense methods under a wide variety of attacks, including white-box and black-box attacks.

Table 1: Performance of our method against white-box attacks on CIFAR-10 dataset ($\epsilon = 8/256$).

| Defense Methods | Clean | FGS | PGD | BIM | C&W |
|---|---|---|---|---|---|
| No Defense | 94.38% | 31.89% | 0.00% | 0.00% | 0.99% |
| Adversarial BIM | 87.00% | 52.00% | – | 32.00% | 42.00% |
| Label Smoothing (Warde-Farley, 2016) | 92.00% | 54.00% | – | 8.00% | 2.00% |
| Feature Squeezing (Xu et al., 2017) | 84.00% | 20.00% | – | 0.00% | 78.00% |
| PixelDefend (Song et al., 2018) | 85.00% | 46.00% | – | 70.00% | 80.00% |
| Adversarial-PGD (Tramr et al., 2018) | 83.50% | 67.92% | 60.15% | – | – |
| Adversarial network (Wang & Yu, 2019) | 91.32% | 73.77% | 49.55% | – | – |
| Our method ($Qs = 5$) | 91.65% | 75.03% | 85.84% | 86.39% | 85.47% |
| Our method ($Qs = 10$) | 91.65% | **75.22%** | **86.05%** | **86.43%** | **86.42%** |

Table 2: BPDA attack results on CIFAR-10 dataset. Results with  are achieved with additional adversarial training.

| Defense Methods | Distance | Accuracy |
|---|---|---|
| TE (Buckman et al., 2018) | 0.031 ($L_\infty$) | 0%[*] |
| SAP (Dhillon et al., 2018) | 0.031 ($L_\infty$) | 0% |
| LID (Ma et al., 2018) | 0.031 ($L_\infty$) | 5% |
| PixelDefend (Song et al., 2018) | 0.031 ($L_\infty$) | 9%[*] |
| Cascade Adversarial Training (Na et al., 2018) | 0.015 ($L_\infty$) | 15% |
| PGD Adversarial Training (Madry et al., 2018) | 0.031 ($L_\infty$) | 47%[*] |
| STL (Sun et al., 2019) | 0.031 ($L_\infty$) | 42%[*] |
| Our method ($Qs = 5$) | 0.031 ($L_\infty$) | 51% |
| Our method ($Qs = 10$) | 0.031 ($L_\infty$) | **53%** |

### 4.1 EXPERIMENT SETUP

Following existing methods in the literature, we use CIFAR-10 and SVHN (Street View House Number) datasets. The CIFAR-10 dataset consists of 60,000 images in 10 classes, with $32 \times 32$ image size. The Street View House Numbers (SVHN) dataset (Netzer et al., 2011) has about 200K images of street numbers. The attack methods to be considered in this work include FGS (Goodfellow et al., 2014b), PGD (Madry et al., 2018), BIM attack (Kurakin et al., 2016), and C&W attack (Carlini & Wagner, 2017).

### 4.2 RESULTS ON THE CIFAR-10 DATASET

We compare the performance of our defense method with 6 state-of-the-art methods developed in the literature under four different white-box attacks: FGS attack (Goodfellow et al., 2014b), PGD (Madry et al., 2018) attack, BIM attack (Kurakin et al., 2016) and C&W attack (Carlini & Wagner, 2017). Following (Kannan et al., 2018) and (Wang & Yu, 2019), the white-box attackers generate adversarial perturbations within a range of $\epsilon = 8/255$. In addition, we set the step size of attackers to be $\epsilon = 1/255$ with 10 attack iterations as the baseline settings. Table 1 shows image classification

Table 3: Performance of our method against black-box attacks on CIFAR-10 ($\epsilon = 8/256$).

| Defense Methods | No Attack | FGS | PGD |
|---|---|---|---|
| No Defense | 94.38% | 63.21% | 38.71% |
| Adversarial-PGD (Tramr et al., 2018) | 83.50% | 57.73% | 55.72% |
| Adversarial network (Wang & Yu, 2019) | 91.32% | 77.23% | 74.04% |
| Our method ($Qs = 5$) | 91.65% | 77.01% | 77.90% |
| Our method ($Qs = 10$) | 91.65% | **77.56%** | **79.81%** |

Table 4: Performance of our method against attacks on SVHN dataset($\epsilon = 0.05$).

| Defense Methods | No Attack | White-box Attack PGD | Black-box Attack PGD |
|---|---|---|---|
| No Defense | 96.21% | 0.15% | 67.66% |
| M-PGD (Madry et al., 2018) | 96.21% | 44.40% | 55.40% |
| ALP (Kannan et al., 2018) | 96.20% | 46.90% | 56.20% |
| Adversarial-PGD (Tramr et al., 2018) | 87.45% | 42.96% | 83.23% |
| Adversarial network (Wang & Yu, 2019) | 96.21% | 37.97% | 81.68% |
| Our method ($Qs = 5$) | 96.00% | 93.39% | 87.15% |
| Our method ($Qs = 10$) | 96.00% | **93.80%** | **88.69%** |

accuracy with different defense methods on the CIFAR-10 dataset. The second column shows the classification accuracy when the input images are all clean. We can see that some methods, such as the Adversarial BIM, Feature Squeezing, and Adversarial PGD degrade the classification accuracy of clean images. This implies that their defense methods have caused significant damages to the original images, or they cannot accurately tell if the input image is clean or being attacked. The rest four columns list the final image classification accuracy with different defense methods. Some methods did not provide results on specific attack methods, which were left blank (marked with '-') in the table. For all of these four attacks, our methods significantly outperforms existing methods. For example, for the powerful PGD attack, our method outperforms the Adversarial-PGD method by more than 28%. We can also see that the GCLN with quantization step size $Qs = 10$ achieves better efficiency than that with $Qs = 5$. This is because the quantized nonlinear transform layers with larger quantization parameters are relatively more efficient in removing the noise in feature maps.

**Defending against the BPDA attack.** The Backward Pass Differentiable Approximation (BPDA) (Athalye et al., 2018) attack is very challenging to defend since it can iteratively strengthen the adversarial examples using gradient approximation according to the defense mechanism. Table 2 summarizes the defense results of our algorithm in comparison with other seven methods. We can see that our GCLN is able to improve the classification accuracy up the second best method by 6%.

**Defending against black-box attacks.** We generate the black-box adversarial examples using FGS and PGD attacks with a substitute model (Papernot et al., 2017). The substitute model is trained in the same way as the target classifier with ResNet-34 network (He et al., 2016) structure. Table 3 shows the performance of our defense mechanism under back-box attacks on the CIFAR-10 dataset. The adversarial examples are constructed with $\epsilon = 8/256$ under the substitute model. We observe that the target classifier is much less sensitive to adversarial examples generated by FGS and PGD black-box attacks than the white-box ones. But the powerful PGD attack is still able to decrease the overall classification accuracy to a very low level, $38.71\%$. We compare our method with the Adersarial-PGD (Madry et al., 2018) and Adversarial Network (Wang & Yu, 2019) methods. We include these two because they are the only ones that provide performance results on CIFAR-10 with black-box attacks. From the Table 3, we can see our method improves the accuracy by 5.8% over the state-of-the-art Adversarial Network method for the PGD attack.

4.3 RESULTS ON THE SVHN DATASET.

We evaluate our GCLN method on the SVHN dataset with comparison with four state-of-the-art defense methods: M-PGD (Madry et al., 2018), ALP (Kannan et al., 2018), adversarial PGD (Tramr et al., 2018) and Adversaral network (Wang & Yu, 2019). For the SVHN dataset, as in the existing methods (Kannan et al., 2018; Wang & Yu, 2019), we used the Resnet-18 (He et al., 2016) for the target classifier. The average classification accuracy is 96.21%. We use the same parameters as in (Kannan et al., 2018) for the PGD attack with a total magnitude of $\epsilon = 0.05$ (12/255). Within each single step, the perturbation magnitude is set to be $\epsilon = 0.01$ (3/255) and 10 iterative steps are used.

**Defending against white-box attack.** Table 4 summarizes the experimental results and performance comparisons with those four existing defense methods. We can see that on this dataset the PGD attack is able to decrease the overall classification accuracy to an extremely low level, 0.15%.

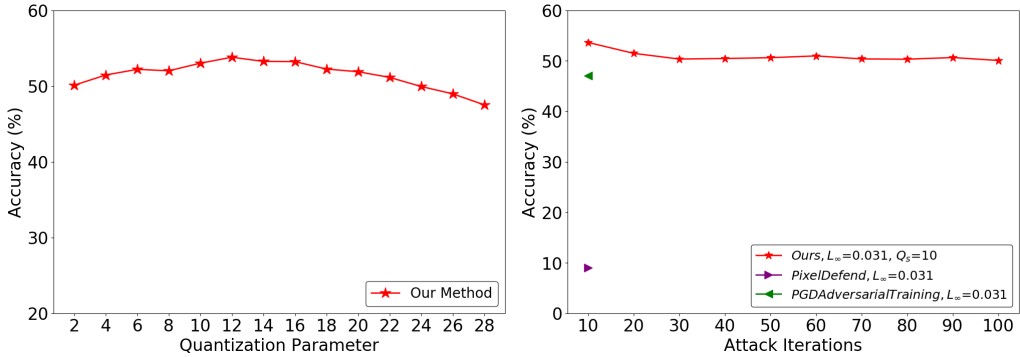

Figure 2: Visualization of the generative cleaning process. The adversarial perturbation was produced using PGD attack with maximum perturbation $\epsilon = 16/255$.

Figure 3: Left: Classification accuracy of our method defense on BPDA attack with different quantization parameter. Right: results against white-box BPDA attack with 10 to 100 attack iterations.

Our algorithm outperforms existing methods by a very large margin. For example, for the PGD attack, our algorithm outperforms the second best ALP (Kannan et al., 2018) algorithm by more than 46%.

**Defending against black-box attack.** We also perform experiments of defending black-box attacks on the SVHN dataset. Table 4 summarizes our experimental results with the powerful PGD attack and provides the comparison with those four methods. We can see that our approach outperforms other methods by 2.39% for the FGS attacks and 7.01% for the PGD attacks. From the above results, we can see that our proposed method is particularly effective for defense against the strong attacks, for example, the PGD attacks with large iteration steps and noise magnitude.

**Visualizing the defense process.** Network defense is essentially a denosing process of the feature maps. To further understand how the the proposed GCLN method works, we visualize the feature maps of original, attacked, and GCLN-cleaned images. We use the feature map from the activation layer, the third from the last layer in the network. Figure 2 shows three examples. In the first example, the first row is the original image (classified into flamigo), its feature map, its gradient-weighted class activation heatmap, and the heatmap overlaied on the original image. The heatmap shows which parts of the original image the classification network is paying attention to. The second row shows the attacked image (being classified into hoopskirt), its feature map, heatmap, and the heatmap overlaid on the attacked image. We can see that the feature map is very noisy and the heatmap is distorted. The third row shows the GCLN-cleaned images. We can see that both the feature map and heatmaps have been largely restored.

## 4.4 ABLATION STUDIES AND ALGORITHM ANALYSIS

In this section, we provide in-depth ablation study results of our algorithm to further understand its capability.

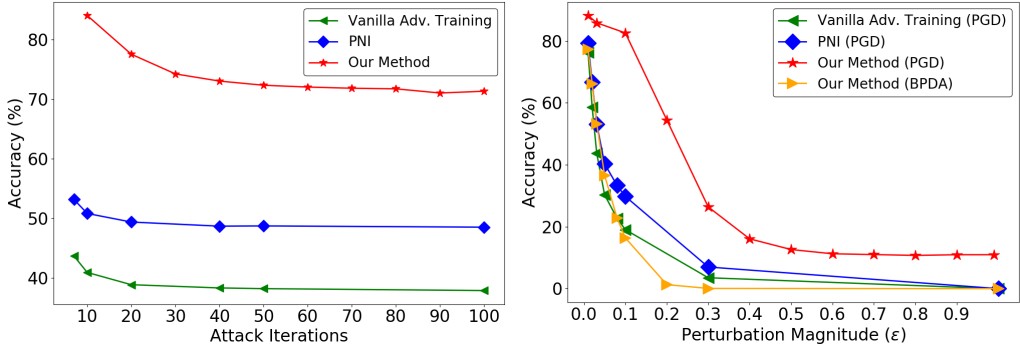

Figure 4: Left: Classification accuracy of our method defense on PGD attack with 10 to 100 attack iterations. Right: results against white-box PGD attack with 0.01 to 1.0 adversarial epsilon.

**(A) Analyze the impact of the quantization parameter.** We notice that the quantization parameter plays an important role in the defense. Figure 3(left) shows the defense performance (classification accuracy after defense) of our method on the CIFAR-10 dataset with white-box BPDA attacks. We can see that the quantization step size within the range of 8 to 12 yields the best performance. Small quantization parameters do not provide efficient defense since the quantized nonlinear transform is not able to disturb and destroy the attack noise pattern. However, when the quantization parameter becomes too large, it will damage the original image content too much which cannot be recovered by the subsequent generative cleaning network.

**(B) Defense against large-iteration BPDA attacks.** The impact of the white-box BPDA attacks increases with its number of iterations since it accesses the network and performs gradient back-propagation with more iterations to force the network towards wrong classification output. Following the protocol of ALP (Kannan et al., 2018), we evaluate the capacity of our defense method against different numbers of BPDA white-box attack iterations. Figure 3(right) shows the performance of our method with an increasing number of attack iteration. We can see that our method is able to withstand large number of BPDA attack iterations. The impact of attack becomes relatively stable after 50 iterations.

**(C) Defense against large-iteration and large-epsilon PGD attacks.** As shown in Fig. 4, gradually increasing the PGD attack iterations will raise the attack strength. This significantly degrades the accuracy of the vanilla adversary training method (Madry et al., 2018) and the PNI (Parametric Noise Injection) method (He et al., 2019), as well as our method. But, our method signficantly outperforms the other two. In both cases, the perturbed-data accuracy starts saturating and do not degrade any further when $N_{step} \geq 40$. Our method still outperforms the vanilla adversarial training and PNI defense methods even when the magnitude of adversarial noise is increased up to $\epsilon = 0.3$.

## 5 CONCLUSION

We have developed a new method for defending deep neural networks against adversarial attacks based on generative cleaning networks with quantized nonlinear transform. This network is able to recover the original image while cleaning up the residual attack noise. We developed a detector network, which serves as the dual network of the target classifier network to be defended, to detect if the image is clean or being attacked. This detector network and the generative cleaning network are jointly trained with adversarial learning so that the detector network cannot find any attack noise in the output image of generative cleaning network. Our extensive experimental results demonstrated that our approach outperforms the state-of-art methods by large margins in both white-box and black-box attacks. For example, it dramatically improves the classification accuracy upon the second best method more than 30% on the SVHN dataset and more than 14% on the challenging CIFAR-10 dataset.

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
