# OpenReview forum: "Generative Cleaning Networks with Quantized Nonlinear Transform  for  Deep Neural Network Defense"
_ICLR.cc/2020/Conference — Reject_

### Official Review · AnonReviewer3 · 2019-10-23
**Official Blind Review #3**

**Rating:** 1

**Review:**

This paper proposes a method for adversarial defense based on generative cleaning.

The paper does not follow any of the best practices for evaluating adversarial robustness, e.g. in these two papers:
"On Evaluating Adversarial Robustness" https://arxiv.org/abs/1902.06705
"Obfuscated Gradients Give a False Sense of Security" https://arxiv.org/abs/1802.00420

For instance the paper does not use a large number of PGD iterations (10 is too small) and does not check that accuracies go to zero for large epsilon (an important sanity check to reveal gradient masking). In the one place where a larger number of attack iterations is used (100 for BPDA) the gap with adversarial training mostly vanishes.

In the absence of these best practices it is impossible to assess the validity of the results, so the paper should be rejected.

**Experience Assessment:**

I have published in this field for several years.

**Review Assessment: Checking Correctness Of Derivations And Theory:**

N/A

**Review Assessment: Checking Correctness Of Experiments:**

I assessed the sensibility of the experiments.

**Review Assessment: Thoroughness In Paper Reading:**

I made a quick assessment of this paper.

---

> ### Author Response · Authors · 2019-11-07
> **Responses to Reviewer 3 comment**
>
> As we can see from our paper, we followed exactly the same evaluation procedure and used the same datasets as the paper mentioned by the reviewer. Following existing papers recently published in ICLR/CVPR/ICCV/ECCV 2017-2019, we used the advTorch standard package to generate all attacks on our method.
>
> Figure 3 shows the large number of BPDA iterations. Due to the page limitation, we did not include the figure for larger number of PGD iterations, since BPDA is a more powerful attack method than PGD.
>
> We could not include the figure for attacks with large epsilon since this figure is not very critical and many recent papers chose not to include it due to page limitations.
>
> Following recently published paper, we have included the most comprehensive performance comparison results in the paper. We have demonstrated  that our method significantly outperforms existing state-of-the-art methods.

---

> > ### Comment · AnonReviewer3 · 2019-11-08
> > **Please clarify re: best pracitces**
> >
> > Hello,
> >
> > I do not understand your response:
> >
> > "As we can see from our paper, we followed exactly the same evaluation procedure and used the same datasets as the paper mentioned by the reviewer."
> >
> > Either I am confused or this is false, as I brought up two explicit best practices you did not follow: a large enough number of PGD iterations, and checking that accuracy goes to zero for large epsilon. These do not seem to be in the paper, if I have missed them please tell me where they are. If you are claiming that you performed them but did not include them in the paper, then please update the paper to include them so that I can look at the numbers and re-assess.
> >
> > "Figure 3 shows the large number of BPDA iterations. Due to the page limitation, we did not include the figure for larger number of PGD iterations, since BPDA is a more powerful attack method than PGD. "
> >
> > BPDA is not uniformly stronger than PGD; it is more finicky, and so while it sometimes works when PGD fails it is important to also use PGD. Moreover it is not clear that 100 iterations is enough for BPDA due to its more finicky nature. In any case, I would like to see PGD vs. number of iterations with number of iterations going at least up to 100 and preferably higher.
> >
> > "We could not include the figure for attacks with large epsilon since this figure is not very critical and many recent papers chose not to include it due to page limitations. "
> >
> > If you are pressed for space I recommend removing the black-box evaluation (since black-box accuracy is mostly meaningless) to make room for more careful checks of the white-box evaluation. But regardless of space limitations, the large epsilon sanity check is essential for assessing the method.

---

> > > ### Author Response · Authors · 2019-11-08
> > > **Responses to Reviewer 3 comment**
> > >
> > > Dear Reviewer, Thank you so much for the quick response and appreciate your valueable suggestion! We are updating the paper to include these two figures, as you suggested. We will upload the paper soon and let you know once it is updated.

---

> > > ### Author Response · Authors · 2019-11-08
> > > **Responses to Reviewer 3 comment**
> > >
> > > Dear Reviewer 3, as suggested by you, we have updated the paper to include the large-iterations and large-epsilon PGD attacks of our method. We also compare our algorithm against PNI and vanilla adv. training method. Please see Figure 4. We can see that our method is able to surve the large-iteration PGD attacks and significantly outperform the other two methods. Also, our method performs much better than the other two methods during large-epsilon attacks with epsilon going to 1, as you suggested. Please review. Thank you!

---

> > > > ### Comment · AnonReviewer3 · 2019-11-12
> > > > **Thanks + comments on figure 4**
> > > >
> > > > Thank you for including these figures. If I understand figure 4 (right panel) correctly, the purported accuracy of your method at epsilon=0.4 is above 15%, and extrapolating to epsilon=0.5 it looks like it should be above 10%. However, at epsilon=0.5 an adversary could map every single image to be uniformly gray (all pixels = 0.5), which gives an upper bound on possible performance of 10% since there are 10 classes. (This is likely a very loose upper bound.) So it is concerning that the proposed method appears to overcome this fundamental limit, and calls into question the reliability of the evaluation. It would be helpful if you could comment on this.

---

> > > > > ### Author Response · Authors · 2019-11-12
> > > > > **Responses to Reviewer 3 comment**
> > > > >
> > > > > We really appreciate your insightful comment!
> > > > >
> > > > > We forgot to plot more points on the curve. Yes, even with epsilon = 0.2, the accuracy with BPDA attack becomes 1.35%, far below 10%, according to our evaluation. However, with PGD at epsilon=0.5, our accuracy is 12.63%, which is slightly above 10% as you mentioned. This is because our nonlinear transform layer has disrupted the gradient propagation behavior of PGD. Because of this, the attacked image by PGD is messed up, instead of being totally random. When we set the epsilon to 0.6, the accuracy drops to near 10%. This shows BPDA is more effective than PGD with our defense. Hope this has addressed your concern. We will update the figure in the paper shortly to include these additional points on the curve, plus the large-epsilon BPDA curve. Thank you very much!

---

> > > > > > ### Comment · AnonReviewer3 · 2019-11-14
> > > > > > **Thanks + question about BPDA comparisons**
> > > > > >
> > > > > > Thanks. It appears then that BPDA at least is needed for accurately evaluating adversarial robustness of your model. My understanding is that Table 2 is the main place where you compare performance against BPDA to other models. Can you clarify how many BPDA iterations was used for this table? Am I missing any other place where the BPDA comparison is performed? In general more info about the implementation of BPDA would be helpful since it is the main relevant metric in the paper.
> > > > > >
> > > > > > I also had a question regarding the training procedure; a contrast you make with other methods is that they perform adversarial training whereas your method does not, but it seems that the generator-discriminator algorithm used at training time is essentially doing adversarial training (but on the generator rather than the final classifier). Is this correct? Or if this is wrong could you explain why?

---

> > > > > > > ### Author Response · Authors · 2019-11-15
> > > > > > > **Responses to Reviewer 3 comment**
> > > > > > >
> > > > > > >
> > > > > > > We really appreciate your valuable comment!
> > > > > > >
> > > > > > > BPDA is an attack method. In the original paper of BPDA, they provided results on MINST, CIFAR-10, and ImageNet. Here is the reason that we only had BPDA defense results on the CIFAR-10. (1) We found that only one defense paper had results on MINST, so we did not provide comparisons on MINST. (2) For the ImageNet, it is too huge and extremely time consuming for us to run all of these experiments. (3) In recently defense papers, only the STL method from CVPR 2019 provided results on BPDA, which we have included it in the paper and our algorithm significantly outperforms it.
> > > > > > > We used the standard attack-defense evaluation package called AdverTorch which provides implementations of major attack methods, including BPDA.
> > > > > > >
> > > > > > > We set 10 attack iterations as the BPDA baseline setting. In addition, the large iterations and large epsilon evaluation of BPDA attacks and are also presented in Figure 3 and Figure 4, which show the proposed method is able to effective defend BPDA attacks and outperform other methods.
> > > > > > >
> > > > > > > We did used the adversarial samples for training the generator and discriminator. But we did not re-train the target classifier and made no modification to the target classifier.

---

### Official Review · AnonReviewer4 · 2019-10-23
**Official Blind Review #4**

**Rating:** 8

**Review:**

This paper proposes a new method to defend a neural network agains adversarial attacks (both white-box and black-box attacks). By jointly training a Generative Cleaning Network with quantized nonlinear transform, and a Detector Network, the proposed cleans the incoming attacked image and correctly classifies its true label. The authors use state-of-the-art attack methods on various models, and the proposed model consistently outperforms all baseline models, even dramatically outperforming them for some specific attack methods.

Comment:
Is there a reason the authors did not test the same set of attack methods for SVHN as they did for CIFAR-10?

**Experience Assessment:**

I do not know much about this area.

**Review Assessment: Checking Correctness Of Derivations And Theory:**

I did not assess the derivations or theory.

**Review Assessment: Checking Correctness Of Experiments:**

I assessed the sensibility of the experiments.

**Review Assessment: Thoroughness In Paper Reading:**

I made a quick assessment of this paper.

---

> ### Author Response · Authors · 2019-11-07
> **Response to Reviewer 4**
>
> We sincerely thank the Reviewer for the positive feedback and high recommendation of our paper!
>
> Thanks for pointing out this! In this paper, we use the benchmark datasets and compare our results against the results published in existing papers.
>
> Unfortunately, not all published papers provided complete results for both of these datasets. For example, some papers did not have results on SVHN. In  this case, we left them empty.
>
> We chose not to re-implement existing methods since it will be very hard to re-produce their exact results, which might lead to unfair performance comparisons.
>
> During our experiments, we tried our best to compare with as many methods as possible if they have published results on the dataset.

---

### Official Review · AnonReviewer1 · 2019-10-28
**Official Blind Review #1**

**Rating:** 3

**Review:**

This paper developed a method for defending deep neural networks against adversarial attacks based on generative cleaning networks with quantized nonlinear transform. The network is claimed to recover the original image while cleaning up the residual attack noise. The authors developed a detector network, which serves as the dual network of the target classifier network to be defended, to detect if the image is clean or being attacked. This detector network and the generative cleaning network are jointly trained with adversarial learning so that the detector network cannot find any attack noise in the output image of generative cleaning network. The experimental results demonstrated that the proposed approach outperforms the state-of-art methods by large margins in both white-box and black-box attacks.

A few comments:

1. It does not provide theoretical reasons why the prosed method can defend against those attacks.

2. The experiments are a bit messy and the attacks' setup need to improve.

3. The proposed defense showed only empirical results against the target attack. It seems to provide no theoretical / provable guarantees.




**Experience Assessment:**

I have published one or two papers in this area.

**Review Assessment: Checking Correctness Of Derivations And Theory:**

I assessed the sensibility of the derivations and theory.

**Review Assessment: Checking Correctness Of Experiments:**

I assessed the sensibility of the experiments.

**Review Assessment: Thoroughness In Paper Reading:**

I read the paper at least twice and used my best judgement in assessing the paper.

---

> ### Author Response · Authors · 2019-11-07
> **Response to Reviewer 1 Comments**
>
> We sincerely thank Reviewer 1 for the positive and encouraging comments!
>
> The reviewer suggests that we need to provide theoretical analysis or proof why the proposed deep neural network defense method is working. We must admit that this is really hard. The deep learning research community is still working very hard to establish theoretical analysis or proof for deep neural networks, which is however is very challenging.
>
> However, our proposed method is based on data-driven deep learning. All the defense networks are trained based on the loss functions. So, if the loss function converges, the proposed method is achieving the target performance.
>
> Following existing state-of-the-art methods published in recent ICLR/CVPR/ICCV/ECCV papers, we use the benchmark their datasets and standard evaluation protocols to demonstrate that our proposed method is significantly outperforming existing methods.
>
> For the experiments, we have results on two datasets, CIFAR-10 and SVHN with two different attack modes: white-box attack and black-box attacks. Sorry for the confusion. We will better organize these experimental results.
>
> For the attacks, we follow the standard procedure used in all existing methods. Specifically, we use advTorch evaluation package to generate all attacks.

---

### Decision · Program_Chairs · 2019-12-19

**Decision:**

Reject

**Comment:**

This paper presents a method to defend neural networks from adversarial attack. The proposed generative cleaning network has a trainable quantization module which is claimed to be able to eliminate adversarial noise and recover the original image.
After the intensive interaction with authors and discussion, one expert reviewer (R3) admitted that the experimental procedure basically makes sense and increased the score to Weak Reject. Yet, R3 is still not satisfied with some details such as the number of BPDA iterations, and more importantly, concludes that the meaningful numbers reported in the paper show only small gains, making the claim of the paper less convincing. As authors seem to have less interest in providing theoretical analysis and support, this issue is critical for decision, and there was no objection from other reviewers. After carefully reading the paper myself, I decided to support the opinion and therefore would like to recommend rejection.